# The Role of Biological Sex in Pre-Clinical (Mouse) mRNA Vaccine Studies

**DOI:** 10.3390/vaccines12030282

**Published:** 2024-03-07

**Authors:** Burcu Binici, Zahra Rattray, Avi Schroeder, Yvonne Perrie

**Affiliations:** 1Strathclyde Institute of Pharmacy and Biomedical Sciences, University of Strathclyde, Glasgow G4 0RE, UK; burcu.eryilmaz@strath.ac.uk (B.B.); zahra.rattray@strath.ac.uk (Z.R.); 2Department of Chemical Engineering, Technion, Israel Institute of Technology, Haifa 32000, Israel; avids@technion.ac.il

**Keywords:** mRNA vaccines, immune response, biological sex, mouse studies, pre-clinical testing, vaccines, LNPs

## Abstract

In this study, we consider the influence of biological sex-specific immune responses on the assessment of mRNA vaccines in pre-clinical murine studies. Recognising the established disparities in immune function attributed to genetic and hormonal differences between individuals of different biological sexes, we compared the mRNA expression and immune responses in mice of both biological sexes after intramuscular injection with mRNA incorporated within lipid nanoparticles. Regarding mRNA expression, no significant difference in protein (luciferase) expression at the injection site was observed between female and male mice following intramuscular administration; however, we found that female BALB/c mice exhibit significantly greater total IgG responses across the concentration range of mRNA lipid nanoparticles (LNPs) in comparison to their male counterparts. This study not only contributes to the scientific understanding of mRNA vaccine evaluation but also emphasizes the importance of considering biological sex in vaccine study designs during pre-clinical evaluation in murine studies.

## 1. Introduction

Both biological sex and gender influence vaccine uptake, responses, and outcomes [1]. A range of studies spanning animal models [2,3] and humans [4,5,6] have reported that biological sex influences immune responses and vaccine efficacy. For instance, Klein et al. demonstrated that female mice, upon vaccination with inactivated influenza A virus, exhibited higher antibody responses and greater activation of germinal centre B cells and memory CD8+ T cells against an influenza challenge compared to their male counterparts [2]. In humans, biological sex-specific responses against the seasonal influenza vaccine have been reported in older adults [7]. Reports also highlight higher adverse effects related to vaccines in females and may reflect a greater immune response in females compared to males [8,9] as could the differences in the prevalence of anti-PEG antibodies in males and females [10]. However, when considering the immune response to COVID-19 infections, the data are less clear, with studies having reported higher immune responses in either females [4,11,12,13] or males [14,15,16], while others found no difference [17,18]. Similarly, whilst the initial reports on the safety and efficacy of two mRNA lipid nanoparticles (LNPs)-based vaccines [19,20] did not fully provide sex-based data [21]. A review considering sex differences in the efficacy of COVID-19 vaccines notes that no statistically significant differences between males and females were reported for the Moderna or Pfizer-BioNTech mRNA vaccines [9].

The factors mediating biological sex-dependent immune responses could be age, sex hormones, and/or genetics-driven, while the factors for gender differences could be behaviour, environment, and/or microbiome when considering sex and gender as a biological variable and social construct, respectively [1,22]. For example, reports have suggested that COVID-19 vaccines can affect the duration and magnitude of the menstrual cycle [23] and mice intravenously injected with mRNA-LNPs during the menstrual cycle demonstrated enhanced LNP accumulation and gene expression in the ovaries and uterus [22]. It has also been reported that elevated levels of estradiol provide a protective effect for young female individuals facing COVID-19, while the decline in estrogen levels in peri-menopausal women (~55 years) is associated with increased fatality rates in women [22,24]. Another factor is the gene dose of Toll-like receptor 7 (TLR-7), one of the pattern recognition receptors for SARS-CoV-2 ssRNA, and females have biallelic TLR7 expression due to having two X chromosomes, leading to the production of more type 1 interferon, resulting in stronger immune responses compared to males [25].

Lipid nanoparticles (LNPs) play a pivotal role in the formulation of mRNA COVID-19 vaccines developed by Pfizer/BioNTech and Moderna. These LNPs are composed of four lipids: a neutral phospholipid (in both vaccines this is distearoylphosphatidylcholine (DSPC)), cholesterol, an ionizable lipid, and a pegylated lipid. Within the Pfizer-BioNTech formulation the ionizable lipid is ALC-0315 ([(4-hydroxybutyl)azanediyl]di(hexane-6,1-diyl)bis(2-hexyldecanoate)) and the pegylated lipid is ALC-0159 (2-[(polyethylene glycol)-2000]-N,N ditetradecylacetamide) and these are at the molar lipid ratio (%) of 9.4:42.7:46.3:1.6 (DSPC–Chol–Ionisable lipid–PEGylated lipid) [26]. Within the Moderna mRNA formulation, the ionizable lipid is SM-102 (heptadecan-9-yl 8-((2-hydroxyethyl) (6-oxo-6-(undecyloxy) hexyl) amino) octanoate) and PEG2000-DMG (1,2-Dimyristoyl-sn-glycerol-3-methoxypolyethylene glycol) is the pegylated lipid. These are at the molar lipid ratio (%) of 10:38.5:50:1.5 [26]. Within these LNPs, the DSPC and cholesterol are added to stabilise the particle. The ionisable lipid is used to complex the mRNA, at low pH these ionisable lipids are cationic and interact with the anionic mRNA during particle formation. The PEG lipid is added to control the particle size and stabilise the particles during storage [26].

Following administration of mRNA-LNP vaccines, a dynamic cellular process ensues where LNPs are taken up by cells through endosomal pathways, facilitating the release of mRNA into the cytosol. The released mRNA is then translated into the corresponding antigen protein within the ribosome, consequently initiating the activation of the immune system. Considering this interplay between mRNA-LNP vaccines and the immune system, it is crucial to explore potential variations in immune responses across different biological sexes. Discrepancies in immune responses after vaccination could be influenced by multiple factors. For instance, depending on the biological sex, LNPs may undergo different processing, encountering unique biological environments associated with each biological sex. Indeed, biological sex-dependent protein corona formation has been reported [27], suggesting that the biological sex of the host could impact the interaction between LNPs and proteins in the body. Any differences in the protein corona may change LNP surface properties, which in turn could impact cellular uptake and the processing of LNPs and, hence, affect biodistribution, clearance, and expression profiles [28,29,30,31].

Given the significance of these factors, our study aimed to investigate biological sex-dependent immune responses by comparing antibody responses and protein expression in female and male BALB/c mice. These mice were intramuscularly injected with mRNA LNPs, typical of a standard vaccine mouse model. The primary objective was to investigate antibody response disparities across the biological sexes and determine whether these differences are driven by mRNA expression profiles or immunological responses. This research seeks to add valuable insights into designing pre-clinical mouse mRNA vaccine studies and to understand the potential impact of biological sex on mRNA-LNP vaccination efficacy.

## 2. Materials and Methods

### 2.1. Materials

The ionizable lipid 8-[(2-hydroxyethyl)[6-oxo-6-(undecyloxy)hexyl]amino]-octanoic acid, 1-octylnonyl ester (SM-102) was purchased from BroadPharm (San Diego, CA, USA). 1,2-distearoyl-sn-glycero-3-phosphocholine (DSPC), 1,2-dimyristoyl-rac-glycero-3-methoxypolyethylene glycol-2000 (DMG-PEG2000) were procured from Avanti Polar Lipids (Alabaster, AL, USA). Cholesterol (Chol), citric acid, sodium citrate tribasic dehydrate, sulfuric acid, hydrogen peroxide (H_2_O_2_), Tween 20, 3,3′,5,5′-Tetramethylbenzidine dihydrochloride hydrate (TMB) were acquired from Merck (Rahway, NJ, USA). EZ Cap™ Firefly Luciferase mRNA (5-moUTP) was procured from APExBIO (Houston, TX, USA). Ovalbumin (OVA)-encoding mRNA modified with 5-methoxyuridine (5moU) (MRNA41) was purchased from OZ Bioscience (Marseille, France). Goat Anti-Mouse IgG (H + L) Secondary Antibody HRP (A16066) was bought from Invitrogen. Goat Anti-Mouse IgG2a-HRP (1081-05) and Goat Anti-Mouse IgG1-HRP (1071-05) were gained from Southern Biotech (Birmingham, AL, USA). Quant-it Ribogreen RNA assay kit, 1,1-dioctadecyl-3,3,3,3-tetramethylindotricarbocyanine iodide (DiR) were bought from Invitrogen (Carlsbad, CA, USA). Minimal Essential Medium (MEM), fetal bovine serum (FBS), sodium pyruvate, and penicillin/streptomycin were acquired from Gibco, Thermo Fisher Scientific (Paisley, UK). Vivo Glo luciferin substrate was acquired from Promega (Southampton, UK). All other solvents were of analytical grade and were supplied in-house.

### 2.2. LNP Preparation

LNPs were prepared using the NanoAssemblr Benchtop from Precision Nanosystems Inc. (Vancouver, BC, Canada). The lipid phase was composed of 6 mg/mL of DSPC–Cholesterol–SM-102–DMG-PEG2k at a molar ratio of 10:38.5:50:1.5%, while the aqueous phase was prepared with 87 µg/mL of mRNA in 50 mM of pH4 citrate buffer, corresponding to a N/P of 6 (the molar ratio of amine groups of the ionizable lipid to that of phosphate groups of mRNA). mRNA encoding ovalbumin (OVA) as a model antigen and mRNA encoding Firefly luciferase (Fluc) as a reporter gene were used in the immunization study and expression study, respectively. DiR (1% mol), a lipophilic fluorescent dye, was included in the organic phase to track the in vivo organ biodistribution of LNPs in the in vivo expression study, using a DIR filter at an excitation level spectrum of 754 nm and emission spectrum of 778 nm. The microfluidic parameter was kept the same at a 3:1 flow rate ratio (FRR; ratios between the aqueous and organic phase) and 12 mL/min total flow rate (TFR). After self-assembling LNPs using microfluidics, they were purified using a spin column (Amicon^®^ Ultra-15 Centrifugal Filter Unit, 100 kDa).

### 2.3. LNP Characterization by Using Dynamic Light Scattering: Particle Size, Polydispersity, and Zeta Potential

The z-average hydrodynamic diameter, polydispersity index (PDI), and zeta potential were assessed using dynamic light scattering using a Zetasizer Ultra (Malvern Panalytical Ltd., Worcestershire, UK). The instrument was equipped with a 633 nm laser and a detection angle of 173°. To measure particle size and PDI, samples were diluted to 0.1 mg/mL lipid concentration with PBS. To measure zeta potential, samples were diluted with ultrapure water. Mean particle size, PDI, and zeta potential are expressed as the mean ± SD.

### 2.4. Entrapment Efficiency

The encapsulation efficiency of LNPs was determined using the Ribogreen Assay. Briefly, 50 µL of the sample was added to the 96-well black plate in the presence and absence of 0.1 *w*/*v*% Triton X-100 to define total mRNA and unencapsulated mRNA, respectively. The plate was incubated at 37 °C for 15 min to allow for LNP disruption to determine the total mRNA concentration. A total of 100 µL of Ribogreen fluorescent dye was added to the wells with 200 × dilution and 500 × dilution for Triton (+) and Triton (−) wells, respectively. The fluorescence intensity was quantified using a GloMax^®^ Discover Microplate Reader at the excitation and emission wavelengths of 480 nm/520 nm. The encapsulation efficiency (%) and recovery (%) were calculated according to the standard curve without and with Triton to quantify non-encapsulated and total mRNA concentration respectively.

### 2.5. In Vivo Studies

All animals were handled in accordance with the UK Home Office Animals Scientific Procedures Act of 1986 (UK project license number PP1650440/personal license number I52241434) in accordance with an internal ethics board.

#### 2.5.1. Biodistribution and In Vivo mRNA Expression Study

To compare responses between male and female mice, two independent studies were conducted with groups of 2 females, 2 males, and 1 non-injected mouse. The 8–10-week-old BALB/c mice were provided by the Biological Procedure Unit at the University of Strathclyde, Glasgow. LNPs were prepared with Fluc mRNA and labelled with DiR with the aim of assessing the expression profile of Fluc mRNA and tracking the biodistribution of LNPs, respectively. Mice imaging was gained using an in vivo imaging system (IVIS Spectrum, Perkin Elmer, Shelton, CT, USA) and Living Image software ^®^4.7.3 was used for image capture and data analysis. Mice were intramuscularly injected in each hind leg with 5 µg of Fluc mRNA LNPs and then were anaesthetized with 3% isoflurane and transferred to the IVIS cabin, maintaining isoflurane level at 2%. Mice were imaged under a DiR filter at an excitation of 754 nm and an emission of 778 nm. Then, they received a subcutaneous (sc) injection of d-luciferin at a dose of 150 mg Luciferin/kg body weight. At 10 min after the sc injection, mice were imaged for bioluminescence imaging in an open filter during the time defined by auto-exposure settings. These imaging sessions with the DiR filter and bioluminescence were repeated 6, 24, 48, and 192 h post-injection of the LNPs. After the last time point, mice were terminated using a schedule 1 method. Image capture and data analysis were carried out using Living Image ^®^4.7.3 software. Average radiant efficiency and total flux were gained by region of interest for fluorescence and bioluminescence, respectively, and normalized with the control mice. Average radiant efficiency and average total flux were calculated for both the fluorescence intensity and bioluminescence measurements.

#### 2.5.2. Immunization Studies

Female and male BALB/c mice, 8–10 weeks old (average of 20 g), were split into 3 groups (1, 2.5, and 5 µg) of 3 mice, which were obtained from the Biological Procedure Unit at the University of Strathclyde, Glasgow. Each group was tested in two independent studies (thus a total dataset of 6 mice per group). In the first study, a control group was also included to confirm no background responses. The mice were immunized two times with 4-week intervals between the two immunizations. Mice were primed with OVA mRNA LNPs intramuscularly at 1, 2.5, or 5 µg per dose, and the blood was collected on day 27. Then, they were boosted with the second matching dose on day 28. Two weeks after the booster dose (day 42), mice were terminated by cardiac puncture and the blood was collected.

#### 2.5.3. Immunological Readouts—Antibody Responses

A direct enzyme-linked immunosorbent assay (ELISA) was used to detect total IgG, IgG1, and IgG2a in the serum. The plates were coated overnight with 100 μL per well of 1 μg/mL OVA protein in 0.1 M carbonate buffer pH 9.6 at 4 °C. The plates were washed with washing buffer (PBS with 0.05% Tween 20) three times and then blocked with 200 µL of 10% *v*/*v* FBS diluted in PBS pH 7.4 for 2 h at room temperature (RT) to eliminate any non-specific binding. During this time, the serum samples were diluted in PBS containing 5% *v*/*v* FBS. The plates were washed 5 times and then 100 μL of diluted serum samples were added to the wells for 1 h at room temperature. After 5 washes in ELISA washing buffer, the plates were incubated for 1 h with 100 μL of horseradish-peroxidase-labelled Goat Anti-Mouse total IgG (1:2500, Invitrogen), Anti-Mouse IgG1 (1:20,000, Southern Biotech), and Anti-Mouse IgG2a (1:5000, Southern Biotech). After incubation, the plates were washed 5 times and 100 μL of 3,3′,5,5′-Tetramethylbenzidine (TMB) substrate was added to each well and incubated in a dark at room temperature for up to 10 min. The reaction was stopped by adding 100 μL of 0.2 M H_2_SO_4_. The absorbance was immediately measured at an optical density of 450 nm (OD450) using an iMark™ Microplate Absorbance Reader (Bio-Rad, Hercules, CA, USA). The endpoint value was defined according to the dilutions and the reciprocal endpoint was calculated.

#### 2.5.4. Statical Information

Data are represented as a means of separate experiments and GraphPad Prism 10 was used to perform statistical analysis by performing ANOVA with post hoc analysis wherever applicable. A *p*-value < 0.05 was considered statistically significant.

## 3. Results

### 3.1. mRNA-LNP Physico-Chemical Attributes

LNPs were prepared with OVA mRNA and Fluc mRNA for the vaccine and protein expression studies, respectively (average molecular weights for OVA mRNA (451 kDa) and FLuc mRNA (672 kDa) are calculated based on the number of nucleotides of 1375 and 1921, respectively). However, given that the LNPs were formulated at N/P of six, this accommodates any differences in the molecular weight of mRNA. The results in Table 1 show that LNPs formulated from DSPC–Cholesterol–SM-102–DMG-PEG2k at a molar ratio of 10:38.5:50:1.5% could effectively entrap both types of mRNA with encapsulation and recovery > 95% in LNPs, which were 69 to 76 nm in z-average diameter, with a low Polydispersity Index (PDI) (<0.02) and near-neutral zeta potential that was not significantly different between the two different mRNA payloads.

### 3.2. Immune Responses in Female and Male Mice after mRNA-LNP Immunisation

BALB/c mice were immunised with 1, 2.5, or 5 µg of OVA mRNA LNPs as characterised in Table 1. Mice were dosed via intramuscular injection with a prime (day 0) and boosters (day 28). The serum was collected via tail bleeding 27 days after the first injection and two weeks after the second injection (day 42), and the antibody endpoint titre was detected by ELISA. When comparing the responses in the two different biological sexes of mice, the results in Figure 1a demonstrate that female mice mounted a significantly (*p* < 0.05) stronger total IgG response than male mice after the primer dose across all three mRNA doses (1, 2.5, or 5 µg), which was up to 14 times higher in female than male mice when 1 µg OVA mRNA encapsulated in LNPs was administered. After a booster dose, total IgG titres increased for all mice and again the female mice mounted significantly (*p* < 0.05) higher IgG responses compared to male mice (Figure 1b). Indeed, the female mice overall had 20-, 8-, and 5-times higher immune responses than male mice immunised with 1 µg, 2.5 µg, or 5 µg of OVA mRNA encapsulated in LNPs, respectively (Figure 1b). When considering IgG dose responses, a linear dose–response association is seen in male mice (R2 ≈ 1) but is not apparent in female mice (Figure 1a,b). With IgG1 (Figure 1c,d) and IgG2a (Figure 1e,f) responses, antibody titres were lower in all mice, more widely spread and with no dose–response relationship. After the first injection, whilst generally there is a trend of higher antibody responses in female mice, significant differences were only noted after the first dose at 1 and 5 µg mRNA doses for IgG1 (Figure 1c,e). After the second (booster) injection, IgG1 and IgG2a responses were similar in female and male mice except for a higher 5 µg mRNA dose, which stimulated a significantly high IgG2a titre in female mice (Figure 1d,f). Across the three doses, the ratio of IgG2a/1 was not significantly different between female and male mice (Figure 1g).

### 3.3. mRNA-Encoded Protein LNPs: Clearance and Expression in Female and Male Mice after Intramuscular Injection

Given female mice had stronger antibody responses compared to male mice when vaccinated with OVA-encoded mRNA-LNPs, we further investigated if these differences in response were the result of differences in mRNA protein expression profiles between the female and male mice or from different immune responses against the encoded protein. To study this, BALB/c mice were intramuscularly injected with DiR-labelled LNPs encapsulating 5 µg of mRNA encoding luciferase (Fluc). By using DiR-labelled mRNA-LNPs, we could track both the biodistribution of LNPs (by imaging the mice under a DIR fluorescence filter) and mRNA luciferase expression (via bioluminescence imaging) [32].

Figure 2 shows the fluorescence intensity profiles for both the male and female mice over the 196 h of the study, with Figure 2a representing IVIS images at 6, 24, and 48 h. Figure 2b shows the distribution profile and Figure 2c shows the Area Under the Curve (AUC). The results show that both males and females have similar clearance rates of the DiR-labelled LNPs from the intramuscular injection site, with peak fluorescence intensity (Cmax) being measured 6 h after mRNA-LNP injection (Figure 2b), with no significant difference in the AUC between female and male mice (Figure 2c).

In line with the distribution data, luciferase expression was primarily focused on the injection site and was not significantly different between female and male mice (Figure 3), with peak luciferase expression measured at the 6 h timepoint (Figure 3a,b). Luciferase expression was also detected in the liver (Figure 3a,c). The results presented in Figure 3c indicated that, at 6 h, there are significantly (*p* < 0.05) higher luciferase expression levels in the liver in female mice; however, at all other time points, no statistically significant difference is observed between the expression levels in the female and male mice. These results confirmed that the difference in immune responses between female and male mice is not due to differences in mRNA-protein expression or LNP distribution after intramuscular injection and is a result of differences between female and male mice in generating immune responses to mRNA-encoded antigens.

## 4. Discussion

In our research, we have confirmed that following intramuscular injection in BALB/c mice, females demonstrate significantly (*p* < 0.05) higher IgG immune responses against the mRNA-encoded antigen (OVA) compared to their male counterparts across the three doses tested (Figure 1). Additionally, we observed comparable mRNA-LNP clearance from the injection site (Figure 2) and protein (luciferase) expression at the injection site for both female and male mice but females did exhibit higher luciferase expression within the liver (Figure 3). However, it is important to recognise that the inherent properties and functions of the expressed protein (OVA and Fluc) could potentially influence various aspects of antigen trafficking, processing, and subsequent immune responses. A study investigating expression kinetics and immunogenicity of mRNA-LNPs considered the correlation between luciferase expression and anti-luciferase responses using a range of LNP formulations, including SM-102 LNPs, like the formulation used in our study. The authors note that LNPs that induced the highest total luciferase expression induced significant anti-luciferase serum antibody titres when compared to empty particles [33]; however, it is also acknowledged that luciferase is reported to be a relatively weak immunogen and it may not reflect the efficacy of other vaccine antigens [34]. Nonetheless, these disparities highlight the intricate interplay between biological sex and immune responses, emphasising the importance of considering these factors when designing pre-clinical mRNA-LNP vaccine studies.

The effectiveness of mRNA-encoding therapeutic proteins and vaccines is influenced by factors such as the administration route and the location, duration, and magnitude of protein production. Depending on the injection site, both localised protein expression and disseminated expression in the liver can occur [35]. Moreover, following intramuscular administration, the depth of the injection can determine differential expression levels at the muscle versus the liver. Deep intramuscular injections of mRNA-LNPs lead to strong protein production in the liver, compared to superficial injections which predominately yield protein in the muscle [35]. Whilst we did not see discernible differences in protein expression levels in the muscle between female and male mice, variations in muscle density could potentially contribute to differences in luciferase expression measured in the liver (Figure 3). The mRNA-LNP luciferase location within the mouse tissue/body will also impact the strength of the fluorescent and luminescent signals. Indeed, detecting the fluorescence intensity in internal organs with IVIS is challenging without organ extractions due to the limited penetration depth in tissue, explaining why we detect mRNA-LNPs at the injection site and not at the liver (Figure 2).

When evaluating vaccine efficacy, numerous studies—including those in murine and human models—have previously highlighted the influence of biological sex on immunological responses [15,36], as we report in Figure 1. For instance, Breznik et al. explored the impact of biological sex and the female reproductive cycle on circulating leukocyte levels in C57BL/6J mice. Their investigation revealed that while biological sex significantly shapes the prevalence and variability of immune cells in peripheral blood, there is no substantial effect associated with the female reproductive cycle [37]. Another study showed that women elicited a greater immune response to inactivated influenza vaccination in women of all ages, regardless of dose or influenza strain, and women had an equivalent antibody response with half the dose of influenza vaccine compared to men [38]. Sex differences in immune responses to respiratory viruses have also been observed in both humans and experimental rodent models [39]. The influence of sex hormones on these responses may involve direct actions on innate immune cells or indirect modulation by other immune or non-immune cells responding to sex hormones [39]. The influence of hormones, specifically testosterone and oestrogens, has been shown to play a role in modulating immunological responses, leading to variations in adaptive immunity, T-lymphocyte function, and helper-inducer cell activity [40,41]. It has also been observed that male hormones promote a tendency towards cell-mediated immune responses, while female hormones tend to promote humoral immunity [42]. Within our studies, we tested IgG1 and IgG2a subclasses in addition to IgG (Figure 1), and both female and male mice had higher IgG1 than IgG2a titres, indicating a stronger T helper type 2 (Th2) immune response was activated upon OVA mRNA LNPs vaccination. This is perhaps not surprising as BALB/c mice have been reported to be biased towards Th2 immune responses [43].

After intramuscular injection, mRNA-LNPs can be taken up by tissue-resident dendritic cells and macrophages, eliciting a local immune response at the injection site [44]. Once taken up, mRNA needs to be released from both the endosome and their LNP carrier into the cytosol. Once in the cytosol, the mRNA is translated into the corresponding antigenic protein in the ribosome. Subsequently, this antigen undergoes degradation in the cytosolic proteasomes, revealing antigenic epitopes that form complexes with major histocompatibility complex (MHC) class I. These activated dendritic cells present the antigen to T cells, which become activated and differentiate into cytotoxic T cells or helper T cells [45]. Myocyte transfection by mRNA vaccines can also activate bone-marrow-derived dendritic cells (DCs), contributing to T cell priming [46]. Based on this mechanism, this supports our finding that there is no major biological difference in the expression profiles for mRNA-encoded protein, rather the differences become apparent during the down-stream processing of the antigen to create the immune responses.

Within research, there is a growing acceptance of the need for males and females to be equally represented. This includes pre-clinical research. Indeed, using both sexes—in this context defined by a set of biological attributes—when designing research experiments is now the default for many grant funders and this is long overdue. Previously, using male models in research was a default approach and male animals were used six times more often than females [47]. However, anecdotally, female mice are often used in pre-clinical vaccine studies due to lower levels of aggression towards each other and, thus, are easier to be socially housed in stable groups with compatible cage mates in line with NC3R guidance. Through our investigations, we show that the biological sex of mice does not impact mRNA-LNP protein expression at the injection site but does impact mRNA-LNP vaccine immune responses and, thus, in designing studies this is an important factor. Nonetheless, it is crucial to consider the specific research question and the contextual nuances of the study when designing pre-clinical investigations. Whilst using a mix of male and female subjects to test vaccine safety and efficacy is key, consideration of data handling is also important. For instance, when evaluating the impact of formulation on vaccine efficacy, averaging responses from a mix of male and female mice may lead to a broad data spread, as highlighted in Figure 4; when the IgG responses are pooled, higher variation in the data is noted (Figure 4a). In contrast, in Figure 4b,c, where no significant differences between males and females were observed, the variability is low. Thus, the pooling of data from both sexes might obscure important distinctions in responses to formulations. Equally, designing studies with appropriate mouse numbers and study replicates using both male and female mice may result in higher animal usage than needed to address the research question, particularly when considering the limitations of the mouse model in vaccine studies generally. In such instances, it is important to report the sex and age of the animals used in pre-clinical experiments.

## 5. Conclusions

When testing mRNA-LNP vaccine (OVA) responses, female BALB/c mice showed a stronger immune response than male mice, whilst mRNA-LNP protein (luciferase) expression was not notably different. Given these immune response differences, consideration should be given to this in experimental study plans and the averaging of data across different biological sexes should be discouraged. In all studies, the sex and age of the animals used in pre-clinical experiments should be reported.

## Figures and Tables

**Figure 1 vaccines-12-00282-f001:**
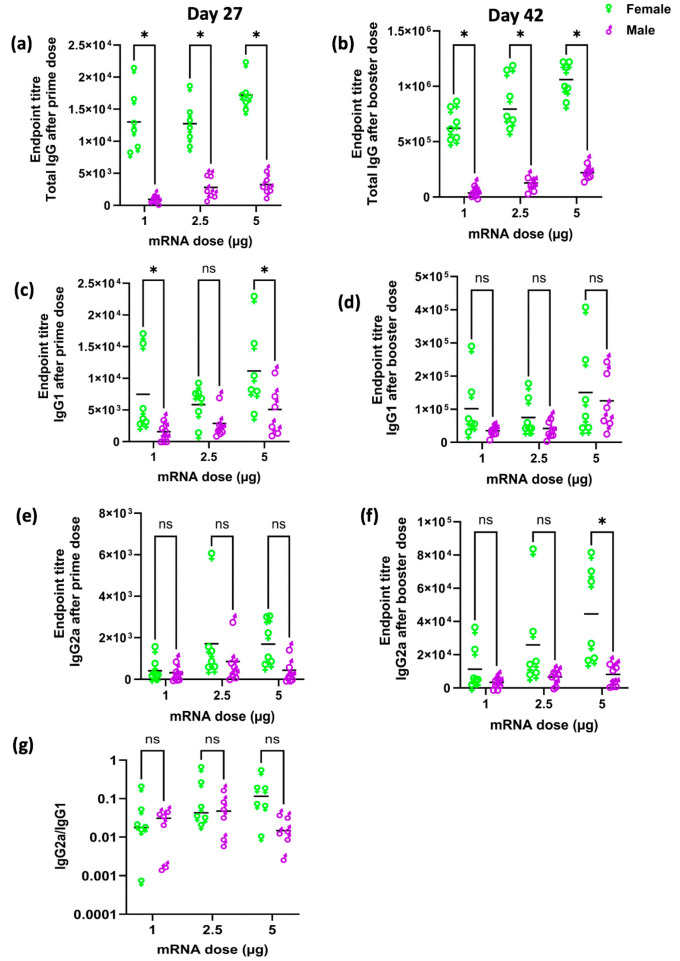
Antibody responses in female and male BALB/c mice immunized with 1, 2.5, or 5 µg of OVA mRNA encapsulated in LNPs on days 0 and 28. Serums were collected after the first and second doses via tail bleeding. Semi-quantitative ELISA was performed. Total anti-OVA IgG endpoint titres elicited by OVA mRNA LNPs after (**a**) first and (**b**) second dose. Anti-OVA protein-directed IgG1 after (**c**) first and (**d**) second dose. Endpoint titter of IgG2a to OVA mRNA LNPs after (**e**) prime, (**f**) booster dose, and (**g**) showing the IgG2a/1 ratio. Each point represents an individual mouse and the black line represents the mean of each group. A total of 18 female mice and 18 male mice (split over 2 independent studies) were included in the statistical analysis performed by GraphPad Prism (ns = not significant; * *p* < 0.05).

**Figure 2 vaccines-12-00282-f002:**
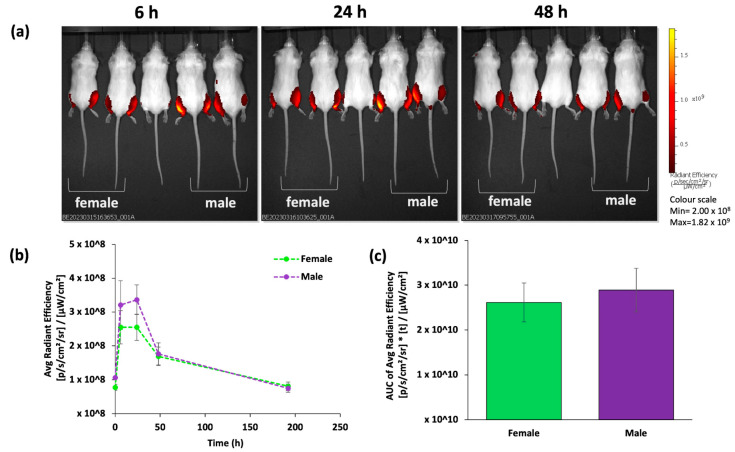
mRNA-LNP fluorescence intensity after intramuscular injection in female and male mice. The injected mRNA dose was 5 µg mRNA encapsulated in LNPs. (**a**) Representative IVIS images at selected time points after DiR-labelled mRNA-LNP injection (left 2 mice are female; right 2 mice are male; middle mouse is not injected). (**b**) Fluorescence signal at the injection site over 196 h and (**c**) the Area Under the Curve (AUC). Data are expressed as mean ± SEM (a total of 4 female mice and 4 male mice split over 2 independent studies).

**Figure 3 vaccines-12-00282-f003:**
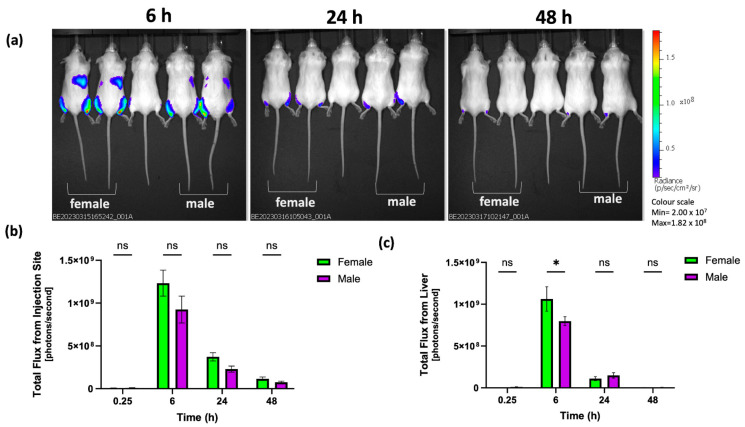
Luciferase expression after Fluc-mRNA LNP intramuscular injection in female and male BALB/c mice. The injected mRNA dose was 5 µg mRNA encapsulated in LNPs. (**a**) Representative bioluminescence IVIS images at selected time points after mRNA-LNP injection. (**b**) Bioluminescence signal per injection site at 0.25, 6, 24 and 48 h and (**c**) Bioluminescence signal per liver at 0.25, 6, 24 and 48 h. Data are expressed by mean ± SEM (a total of 4 female mice and 4 male mice split over 2 independent studies) and statistical analysis was performed by GraphPad Prism (ns = not significant; * *p* < 0.05).

**Figure 4 vaccines-12-00282-f004:**
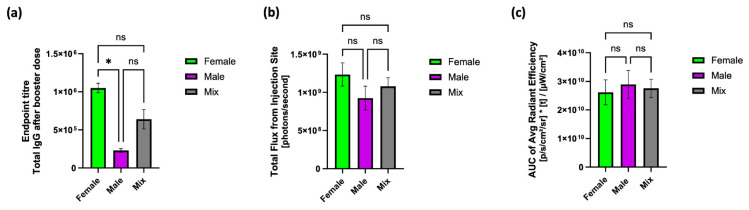
Comparison of responses from female mice, male mice, and the average across all mice for (**a**) IgG responses, (**b**) luciferase expression, and (**c**) AUC. Mice were intramuscularly injected with 5 µg mRNA encapsulated in LNPs with mRNA encoded with either OVA ((**a**) vaccine study) or Fluc ((**b**) expression studies) and labelled with DiR for (**c**) investigating clearance. Data are expressed as mean ± SEM (a total of 18 female mice and 18 male mice split over 2 independent studies for (**a**) and a total of 4 female mice and 4 male mice split over 2 independent studies for (**b**,**c**)) and statistical analysis was performed by GraphPad Prism (ns = not significant; * *p* < 0.05).

**Table 1 vaccines-12-00282-t001:** mRNA-LNPs physico-chemical attributes prepared with different mRNA payloads. Results are expressed as the mean ± SD, n = 3.

	LNP Physico-Chemical Attributes	
mRNA Payload	Ovalbumin (OVA)-Encoding mRNA Modified with 5-Methoxyuridine (5moU)	EZ Cap™ Firefly Luciferase mRNA (5-moUTP)
z-average diameter (nm)	69 ± 6	76 ± 1
PDI	0.04 ± 0.03	0.04 ± 0.01
Zeta Potential (mV)	−1.6 ± 2	−4.5 ± 3
mRNA Encapsulation (%)	95 ± 2	99 ± 1
mRNA Recovery (%)	102 ± 12	106 ± 3

## Data Availability

The supporting dataset can be found at https://doi.org/10.15129/3b4fac4b-84d0-4225-afdd-1e4ef5acf7ca.

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
