# Peer review of "The Role of Biological Sex in Pre-Clinical (Mouse) mRNA Vaccine Studies"

_vaccines, 2024, doi:10.3390/vaccines12030282_

Round 1

Reviewer 1 Report

Comments and Suggestions for Authors

Since the COVID-19 pandemic, the research on mRNA-LNP has taken into the new avenues.  Binici et al. studied the “The Role of Biological Sex in Pre-Clinical (Mouse) mRNA Vaccine Studies”. The authors evaluated the mRNA-LNP immune response in both male and female BALB/c mice. Although there is no significant difference in protein expression, the female mice have shown higher IgG responses than males. 

This reviewer has the following questions/suggestions.

Lines 30-31: Is this statement related to mRNA vaccine response? All cited references are discussed about either SARS-CoV-2 infection or COVID-19 symptoms.

Lines 54-57: More description is needed. Four lipids function is not discussed properly.

Line 149: How much volume was injected through the IM route? We can see the luciferase expression on two sides of the mice, does it mean injections were given to two thigh muscles? please clearly mention this information in the methods section. 

Line 153: "appropriate amount".... please mention the concentration precisely, which is important for reproducibility.

Table 1: Zeta potential looks strange. The authors have not changed the lipids composition then why does the Zeta potential vary with mRNA?

Lines 207-213: This is already told in the materials and methods section. Redundant here.

Figure 1: IgG2a/IgG1 difference should be calculated and presented for all the groups.

Line 285-286: Are these sentences correct? " b) Bioluminescence 285 signal at the injection site over 48 h and c) Bioluminescence signal at the liver over 48 h".  It’s not only 48 h.

Lines 337-335: The authors have tried to give the basics of the vaccine-induced immune response. However, there are serious errors in the context. I would rather do some reading and restructure this paragraph.

For e.g., "These complexes are then presented to APCs, such as CTLs". I'm afraid that's not right. APCs present the antigen completes to T cells.

Second "a portion of the administered mRNA vaccine drains into LNs through the lymphatic system. Within the LNs, monocytes, naive T cells, and B cells are present." The antigen-experienced T cells travel to LNs and generate memory T cells and help in antigen-specific B cell generation, including plasma and memory B cells.

Several errors are present in this paragraph.

The authors should have taken care of references format. A few references are not in proper format, either they are missing journal names or volume numbers or issue or page numbers. Please consider correcting them.

Author Response

Since the COVID-19 pandemic, the research on mRNA-LNP has taken into the new avenues.  Binici et al. studied the “The Role of Biological Sex in Pre-Clinical (Mouse) mRNA Vaccine Studies”. The authors evaluated the mRNA-LNP immune response in both male and female BALB/c mice. Although there is no significant difference in protein expression, the female mice have shown higher IgG responses than males. 

This reviewer has the following questions/suggestions.

Lines 30-31: Is this statement related to mRNA vaccine response? All cited references are discussed about either SARS-CoV-2 infection or COVID-19 symptoms.

Many thanks for raising this point, we have revised the introduction to better explain data and included a new reference that considers a review of mRNA vaccine responses and biological sex (Jensen A, Stromme M, Moyassari S, Chadha AS, Tartaglia MC, Szoeke C, Ferretti MT. COVID-19 vaccines: Considering sex differences in efficacy and safety. Contemp Clin Trials. 2022 Apr;115:106700. doi: 10.1016/j.cct.2022.106700) (see revised manuscript tracked changes).

Lines 54-57: More description is needed. Four lipids function is not discussed properly.

We have added additional text outlining the composition of the 2 mRNA vaccines and the role of each lipid in more detail and we have added the reference to support this (see revised manuscript; tracked changes).

Line 149: How much volume was injected through the IM route? We can see the luciferase expression on two sides of the mice, does it mean injections were given to two thigh muscles? please clearly mention this information in the methods section. 

Mice received an injection into both thigh muscles, this has now been clarified in the revised manuscript (tracked changes).

Line 153: "appropriate amount".... please mention the concentration precisely, which is important for reproducibility.

This has been corrected to include the dose (150 mg Luciferin/kg body weight) (tracked changes).

Table 1: Zeta potential looks strange. The authors have not changed the lipids composition then why does the Zeta potential vary with mRNA?

We have double checked this, there is no significant difference in the zeta potential for the two difference formulations, they are both near neutral. To address this, we have confirmed this in the manuscript (tracked changes).

Lines 207-213: This is already told in the materials and methods section. Redundant here.

This has been deleted.

Figure 1: IgG2a/IgG1 difference should be calculated and presented for all the groups.

Many thanks for the suggestion, we have added an additional graph (Figure 1g) to the manuscript showing the ratio of IgG2a/IgG1.

Line 285-286: Are these sentences correct? " b) Bioluminescence 285 signal at the injection site over 48 h and c) Bioluminescence signal at the liver over 48 h".  It’s not only 48 h.

Thanks for spotting this. We have corrected this now to state the 4 times points included in the graphs (at 0.25, 6, 24 and 48 h).

Lines 337-335: The authors have tried to give the basics of the vaccine-induced immune response. However, there are serious errors in the context. I would rather do some reading and restructure this paragraph.

For e.g., "These complexes are then presented to APCs, such as CTLs". I'm afraid that's not right. APCs present the antigen completes to T cells.

Second "a portion of the administered mRNA vaccine drains into LNs through the lymphatic system. Within the LNs, monocytes, naive T cells, and B cells are present." The antigen-experienced T cells travel to LNs and generate memory T cells and help in antigen-specific B cell generation, including plasma and memory B cells.

Several errors are present in this paragraph.

Many thanks for your guidance and input here, we have corrected this paragraph based on your advice (please see revised manuscript; tracked changes).

The authors should have taken care of references format. A few references are not in proper format, either they are missing journal names or volume numbers or issue or page numbers. Please consider correcting them.

These have been corrected (please see revised manuscript; tracked changes).

Reviewer 2 Report

Comments and Suggestions for Authors

This was a well planned and executed piece of work. I enjoyed reading it and it certainly got me thinking about the way in which murine vaccine models are designed. We also use all female mice in our studies. Also, it is clear that the immune responses are not due to differences in expression levels after vaccination, another important point. Figure 4 of the manuscript summarises the work nicely.

I recommend the manuscript be accepted in its present form.

Author Response

Many thanks, we are glad you find this publication useful.

Reviewer 3 Report

Comments and Suggestions for Authors

The authors submit a very well-written manuscript describing the use of an mRNA vaccine in male and female Balb/c mice. The authors conclude that there are distinct differences in how female mice respond to this vaccine compared to male mice. In particular total IgG responses are increased in female mice. The experiments are well executed with good statistical power. Although these findings are not surprising and have been documented elsewhere the use of mRNA lipid nanoparticles and these documented differences may be of interest to those developing LNP vaccines for clinical use. I have very few concerns with the manuscript. Below are the concerns that should be addressed prior to publication. 

1. The only issue I have with this study is that the authors assume that the OVA-encoded mRNA-LNPs express the OVA protein in a manner that is equivalent to that observed with the Fluc mRNA-LNPs. The Fluc is used as a surrogate for OVA to allow detection, however the properties and function of the proteins themselves can have impacts on antigen trafficking and processing and potentially immune responses downstream. A paragraph in the discussion should be added to address this point. 

2. In line 204 PDI should be spelled out and on Line 211 it should be abbreviated. 

3.  In the Figure 2 legend the authors provide "ns=not significant" but this is not labeled on the figure. 

4. 

Author Response

The authors submit a very well-written manuscript describing the use of an mRNA vaccine in male and female Balb/c mice. The authors conclude that there are distinct differences in how female mice respond to this vaccine compared to male mice. In particular total IgG responses are increased in female mice. The experiments are well executed with good statistical power. Although these findings are not surprising and have been documented elsewhere the use of mRNA lipid nanoparticles and these documented differences may be of interest to those developing LNP vaccines for clinical use. I have very few concerns with the manuscript. Below are the concerns that should be addressed prior to publication. 

  1. The only issue I have with this study is that the authors assume that the OVA-encoded mRNA-LNPs express the OVA protein in a manner that is equivalent to that observed with the Fluc mRNA-LNPs. The Fluc is used as a surrogate for OVA to allow detection, however the properties and function of the proteins themselves can have impacts on antigen trafficking and processing and potentially immune responses downstream. A paragraph in the discussion should be added to address this point. 

Thanks for raising this point; it is a good point we have not covered. We have now included this within the paper, please see revised manuscript (tracked changes).

  1. In line 204 PDI should be spelled out and on Line 211 it should be abbreviated. 

Many thanks for spotting this. We have corrected this in the manuscript.

  1. In the Figure 2 legend the authors provide "ns=not significant" but this is not labeled on the figure. 

Many thanks for spotting this. We have corrected the figure legend to remove both the reference to ns and *; within the text, we clarify there is no significant difference in the AUC (Figure 2c).

Round 2

Reviewer 1 Report

Comments and Suggestions for Authors

The authors have revised the manuscript by considering the reviewer's suggestions. I hereby accept the manuscript for publication in its revised form.